# A Novel Piezoelectric Energy Harvester for Earcanal Dynamic Motion Exploitation Using a Bistable Resonator Cycled by Coupled Hydraulic Valves Made of Collapsed Flexible Tubes

**DOI:** 10.3390/mi15030415

**Published:** 2024-03-20

**Authors:** Tigran Avetissian, Fabien Formosa, Adrien Badel, Aidin Delnavaz, Jérémie Voix

**Affiliations:** 1Université du Québec-École de Technologie Supérieure, Montréal, QC H3C 1K3, Canada; cc-aidin.delnavaz@etsmt.ca (A.D.); jeremie.voix@etsmtl.ca (J.V.); 2Laboratoire SYMME, Université Savoie Mont Blanc, 74940 Annecy, France; fabien.formosa@univ-smb.fr (F.F.); adrien.badel@univ-smb.fr (A.B.)

**Keywords:** energy harvesting, frequency-up, earcanal, multiphysics, hydraulic valve

## Abstract

Scavenging energy from the earcanal’s dynamic motion during jaw movements may be a practical way to enhance the battery autonomy of hearing aids. The main challenge is optimizing the amount of energy extracted while working with soft human tissues and the earcanal’s restricted volume. This paper proposes a new energy harvester concept: a liquid-filled earplug which transfers energy outside the earcanal to a generator. The latter is composed of a hydraulic amplifier, two hydraulic cylinders that actuate a bistable resonator to raise the source frequency while driving an amplified piezoelectric transducer to generate electricity. The cycling of the resonator is achieved using two innovative flexible hydraulic valves based on the buckling of flexible tubes. A multiphysics-coupled model is established to determine the system operation requirements and to evaluate its theoretical performances. This model exhibits a theoretical energy conversion efficiency of 85%. The electromechanical performance of the resonator coupled to the piezoelectric transducer and the hydraulic behavior of the valves are experimentally investigated. The global model was updated using the experimental data to improve its predictability toward further optimization of the design. Moreover, the energy losses are identified to enhance the entire proposed design and improve the experimental energy conversion efficiency to 26%.

## 1. Introduction

The growing use of wireless devices and the miniaturization of electronic circuits have led to significant progress in the energy consumption of wearable mobile devices. Energy harvesting methods have been studied to complement and enhance the power supply and autonomy of batteries. In-ear devices such as hearing aids and cochlear implants are powered by disposable cells or rechargeable batteries. The energy consumption of the best integrated devices in the literature reaches up to 17 J for a 10 h period of daily use [1,2,3]. Woodruff et al. underlined that patients using hearing aids sometimes struggle to change and select disposable batteries for their devices [4]. Another statistical study recently revealed that the rechargeable batteries have the most positive impact on customer’s listening experience [5]. Indeed, the long-term lifespan of rechargeable solutions makes them more economical and ecological compared to disposable cells. Such considerations have spurred the development of energy-harvesting systems.

The common sources of renewable energy are solar or wind, but their use is unfeasible for the scale of this application. Another avenue comes from the fact that the human body generates a considerable amount of energy from various types of movements or conditions. Any of these, the electrochemical energy from the inner ear [6], kinetic energy from walking or head movements [7,8], thermal energy from body heat [9], and strain energy from skin deformation [10] could be harvested for in-ear applications. An interesting energy source for cochlear implants or hearing aids appears to be the earcanal’s mechanical deformation.

Recent studies focusing on energy harvesting for wearables have demonstrated significant potential for developing miniaturized and efficient bio-integrated devices [11]. However, the topic of harvesting energy inside the earcanal is not extensively covered in the existing literature. Most energy harvesters designed for hearing devices are primarily implantable solutions. This requirement imposes the use of flexible materials to accommodate soft human tissue. Latif et al. have recently conducted a review on flexible piezoelectric bio-integrated solutions specifically designed for energy harvesting in implantable devices [12]. The output power of the devices listed ranges from 
$10^{-9}$
 W to 
$10^{-5}$
 W.

While these implantable solutions are compatible with cochlear implants, they may not be suitable for non-invasive hearing devices such as hearing aids or hearables (communication earpieces, Bluetooth earphones, etc.).

In 2012, Delnavaz et al. first studied the power capability of the earcanal’s geometry variation resulting from jaw movements [13]. Carioli et al. then used a customized earplug molding technique with a 3D scanner to characterize the global bending movement and local compression area in the earcanal during mastication [14]. These two mechanical deformations could potentially be exploited as an energy source.

Delnavaz et al. developed a flexible piezoelectric Energy Harvester (EH) made of a silicon earplug with a PVDF layer [15]. The earplug was molded to the subject’s earcanal to maximize surface contact. The aim of the EH was to exploit both the flexion and compression energies. The experimental prototype was able to extract 44 µJ from a mastication cycle. The soft materials facilitated the bio-integration of the EH, but the PVDF’s low electromechanical coupling coefficient may have been the cause of poor energy conversion efficiency. A hydro-electromagnetic energy EH was also developed by the same researchers [13]. The energy was extracted using a liquid-filled earplug whose internal volume varied as a result of the radial compression of the earcanal during jaw movements. The earplug was equipped with a magnet in a water column surrounded by a coil. When the volume varied, the magnet would shift, generating energy by electromagnetic induction. The generated energy experimentally amounted to 
0.2
 µJ per mastication cycle. The prototype appeared to be less efficient than the PVDF earplug. This could be explained by the poor power generation of the electromagnetic transduction for a low-frequency energy source (1.57 Hz) and a small-scale application [16,17]. However, in this proposed strategy, the hydraulic transmission will enable the use of more complex and voluminous technological solutions, since the energy will be made available outside the earcanal.

Based on the analysis of the literature, we propose a new solution to optimize the energy conversion efficiency and maximize the harvested energy from the earcanal’s dynamic motion. The next section states the hydraulic energy source characterization and harvesting strategy that was adopted. Section 3 presents the EH and describes its operation principle. Section 4 details the global system modeling and Section 5 presents the numerical simulation of the theoretical behavior of the EH. Then Section 6 exposes the experimental validation performed on the electromechanical converter stage and Section 7 shows the experimental approach for the design of the hydraulic valves for the system cycling. Section 8 presents the experimentally adjusted global EH model and discusses this work’s major contributions and the potential of the EH. This is followed by the conclusion.

## 2. Energy Source and Harvesting Strategy

### 2.1. The Energy Source

Extracting energy from the mechanical deformations of human tissue requires soft flexible materials so that the earcanal deflection energy can be harvested without causing discomfort. However, flexible transducers have a low energy-conversion efficiency because part of the extracted energy is dissipated in the elastic non-active material. Additionally, the low frequency of mastication makes it difficult to use resonant structures that generally admit better conversion efficiencies than low frequency solutions [18]. Thus, the traditional transducing methods are unsuitable for this application.

To date, the best solution proposed in the literature consists of using soft materials, such as the piezoelectric PVDF earplug previously introduced [15]. In fact, the low mechanical impedance of the soft tissue limits the use of more effective rigid piezoelectric materials.

The hydraulic extraction strategy proposed in [13] is less effective than the piezoelectric earplug, but if energy can be transmitted outside the earcanal, the device could benefit from a larger effective space. The latter is essential to the implementation of an impedance adaptation stage and more effective transducing methods that could not otherwise fit directly inside the earcanal’s limited volume. In fact, Marin et al. demonstrated that the power generated by piezoelectric or electromagnetic transducers is proportionally greater, as the effective volume of the transducing material is greater [19]. Thus, the present work is based on exploiting the earcanal’s deformation using a liquid-filled earplug. This extraction method allows us to use the available space in a traditional hearing casing (50 × 20 × 10 mm) placed behind the ear. As an energy source, the earplug is akin to a micro pump characterized by its volume variation 
$V_{ear}$
 and pressure variation 
$p_{ear}$
. To optimize energy extraction, the earplug must fit the earcanal wall snugly. Turcot et al. demonstrated that the initial pressurization must reach a minimum of 14 kPa to achieve a good fit [20]. Bouchard-Roy et al. studied the dynamic pressure variation inside the earplug for different initial pressurization levels [21]. A maximum of 
$\max(\Delta p_{ear})=12$
 kPa of dynamic pressure variation, with 28 kPa of initial pressurization, was noted during mastication for seven subjects. Noting that no discomfort has been mentioned, the earcanal could potentially support a higher pressurization level. Furthermore, the temporal evolution of the volume variation 
$\Delta V_{ear}$
 has been estimated in [13] and the result for a mastication cycle is presented in Figure 1.

The characteristics of the liquid filled earplug’s operating behavior have not been studied yet. Thus, the evolution of the volume variation 
$\Delta V_{ear}$
 is not well known as a function of the pressure variation 
$\Delta p_{ear}$
. Considering the pressure and volume variation levels from [13,21], Equation (Equation 1) approximates an upper limit for the extractable hydraulic energy with the maximum values of 
$\Delta p_{ear}$
 and 
$\Delta V_{ear}$
 that have been recorded, i.e., 
$p_{c}=12$
 kPa and 
${\left(\Delta V_{ear}\right)}_{max}=60$
 mm^3^. 
$p_{c}$
 stands for the comfort pressure by assuming that the user feels discomfort if 
$\Delta p_{ear}>p_{c}$
. The energy estimation presumes that the earplug is a perfect flow rate source, which is a theoretical hypothesis leading to 
0.72
 mJ available from one mastication.

If a person masticates 2200 cycles per day, which is the average according to [22], there would be 
1.6
 J available per day, per ear for three daily meals. Other activities may significantly raise this energy potential.

The Mehl et al. study published in 2007 that involved over 186 men and 210 women determined that the average number of words pronounced per day is 15,934, with a standard deviation (STD) of 7967 [23]. Assuming an average sentence length of 10 words (STD = 5) [24] and considering that the jaw opens and closes at least once per sentence, this results in an additional 1600 mastication cycles.

Additionally, chewing something between meals can add a significant amount of energy. Noting that the average chewing gum flavor lasts for 15 min [25] and assuming a chewing frequency of 
1.57
 Hz, a single chewing gum use adds 1400 mastication cycles per day.

To summarize, the number of jaw opening and closing cycles per day can reach a significant count of 5200 (2200 for meals, 1600 for speaking, and 1400 for chewing a single piece of gum). It is worth noting that chewing gum alone can replace a typical day of verbal interactions. Additionally, a total of 4 J of energy is available per day per ear. With the two ears, this amount represents 47% of the energy demand for state-of-the-art cochlear implants, which is 17 J for 10 h of daily use [3].

This first estimation shows a promising energy level compared to the supply need. Additionally, the energy source has not been fully characterized yet, suggesting that the earcanal may support higher pressurization levels. Furthermore, the energy consumption of the hearing aids and cochlear implants decreases with time. The continuous improvements in both domains related to hearing aids’ power consumption and energy harvesting efficiency could lead to fully autonomous devices in the future.

(1)
$$\max\left(E_{hyd}\right)=p_{c}\;\cdot \;{\left(\Delta V_{ear}\right)}_{max}=0.72\;\mathrm{mJ}$$


### 2.2. The Energy-Harvesting Strategy

#### 2.2.1. Transducing Method

Electromagnetic transducers are usually used to harvest energy from kinetic (vibration) energy sources. In fact, the power density of electromagnetic transducers decreases proportionately to the effective volume of the transducing material and the frequency of the energy sources decreases [16,17]. Also, the fabrication and integration of the coil and moving magnet may be a source of difficulty, as their gap should be carefully managed to prevent non-linear and unpredicted responses [26].

Piezoelectric energy harvesting has been extensively studied, and numerous efforts have been made to effectively optimize these transducers. Ambient energy sources typically exhibit low frequencies or even aperiodic characteristics. Addressing this issue can be approached in various ways, depending on the specific application. One approach involves adjusting the system resonance through the incorporation of additional piezoelectric, electrostatic, or electromagnetic systems with negative stiffness [27]. Another option is to modify the geometric or mass parameters [28,29]. Some designs utilize multi-resonance frequency systems with different subsystems or systems with a multimodal frequency response [30]. In certain cases, self-coupled structures with mechanical stops are employed to expand the overall resonance frequency range [31]. Additionally, active frequency tuning for broad bandwidth energy harvesting has also been proposed [32].

Extensive research has also been conducted on enhancing the electromechanical coupling of piezoelectric harvesters, as the power generated is directly influenced by it [33]. One straightforward method to significantly improve harvesting performance is by utilizing piezoelectric elements in their 33 mode rather than the 31 mode [34]. Additionally, tuning the geometrical and mass parameters of the system can result in highly coupled systems [35]. Alternatively, exploring bio-inspired geometries has shown promise in achieving interesting harvesting performances [36].

The best energy conversion efficiency for piezoelectric ceramics is achieved at the transducer’s resonance and for strongly coupled electromechanical configurations. As the mean mastication frequency is estimated to be 
1.57
 Hz, the harvesting system would benefit from a frequency up-conversion stage to increase the frequency of the energy source, which in turn increases the energy conversion efficiency of the transducer [18,37]. In frequency up-converting architectures, a high-frequency transducer (HFT) is excited by a low-frequency absorber (LFA). Piezoelectric materials, mechanical stops or escapement mechanisms are often utilized as the LFA, while cantilevers serve as the HFT. This arrangement allows for high frequencies to be achieved while maintaining a strong mechanical coupling [38,39,40,41]. Furthermore, alternative approaches suggest contact-free LFAs composed of permanent magnets, aiming to minimize the dissipative energy caused by stoppers [42,43]. Recent research efforts have focused on the development and optimization of monolithic bistable structures that integrate stacked piezoelectric ceramics within flextensional elastic structures [44]. These structures demonstrate improved performances attributed to their high electromechanical coupling.

In theory, a BR incorporating piezoelectric ceramics within a flextensional structure has the potential to achieve higher energy conversion efficiency compared to low-frequency solutions and/or flexible electromechanical transducers [37,45]. Our work focused on miniaturizing and integrating bi-directional low-frequency excitation into such a solution. Additionally, the design of thin beams was optimized using both analytical and finite element models to maximize the resulting electromechanical coupling.

#### 2.2.2. Power Transmission

The relevance of the hydraulic method for the energy extraction is supported by the efficiency of hydraulic transmission under low fluid inertia. The energy lost by the fluid friction against the tubing wall is directly dependent on the Reynolds number 
Re
 defined by the tubing diameter *D*, the dynamic viscosity 
$\mu_{w}$
 and the density 
ρ
 of the fluid (water). It has been proven that the regular pressure losses are considerable when 
Re>
 (2000–4000), depending on the material of the tubing. Given the experimental data on Figure 1 and the physical properties of water, the regular energy losses in the hydraulic transmission can be minimized if the tubing diameter stays under 
0.5
 mm.

(2)
$$Re=\frac{4\;\rho \;q_{ear}}{\pi \;\mu_{w}\;D}$$


The hydraulic–mechanic interface can be a hydraulic cylinder (HC). It can store the energy in the frequency-up converter by actuating it at the frequency of human mastication. The amount of extractable energy from the hydraulic source will then depend on the EH mechanical behavior. Before being converted into electricity, the energy is in fact stored in a mechanical potential reservoir. During the low-frequency actuation, the forces applied on each side of the HC piston are equal. Equation (Equation 3) models the hydraulic–mechanic interface with 
$x_{m}$
 and 
$S_{hc}$
 representing the dynamic mass position on the 
$\vec{x}$
 axis and the HC internal section.

(3)
$$\Delta p_{ear}\;S_{hc}=K_{s}\;x_{m}$$


Therefore, the potential energy stored in a linear resonator from the ideal hydraulic source previously described can be expressed as follows:
(4)
$$Ep_{lin}=\frac{K_{s}\;x_{m}^{2}}{2}=\frac{\Delta p_{ear}\Delta V_{ear}}{2}$$


Thus, the linear solution can theoretically exploit/store a maximum of 
50%
 from the maximum available hydraulic energy (
$\max(E_{hyd})$
).

Non-linear elastic potential reservoirs can also be considered. This work presents the energy extraction capability of a BR governed by a Duffing equation. It admits two stable equilibrium positions 
$x_{m}=\pm x_{0}$
, with an unstable one in 
$x_{m}=0$
, and its stiffness depends on its structural buckling level 
ϵ
 defined later in Equation (Equation 22). The maximum theoretical energy in the BR for a quasi-static actuation is equal to the height of the potential energy barrier 
$E_{pb}$
 when the mass position varies from 
$x_{m}=\pm x_{0}$
 to 
$x_{m}=0$
 (Equation (Equation 5)). It will be a fraction *n* of the upper limit of the available hydraulic energy (Equation (Equation 1)), where *n* has to be maximized.

(5)
$$E_{pb}=\frac{Kx_{0}^{2}\epsilon^{2}}{3}=n\left(p_{c}\;\cdot \;{\left(\Delta V_{ear}\right)}_{max}\right)$$


Hence, we see that it is inherently capable of exploiting 65% from the hydraulic energy source, which is 15% better than the best theoretical linear option. Section 4 will discuss the modeling and design of a non-linear BR harvester, as this appears to be a better solution than the linear approach.

## 3. Energy Harvester Presentation and Operation Principle

Figure 2 is a schematic view of the energy-harvesting system. In this system, a liquid-filled earplug (Figure 3) transmits the energy outside the earcanal. To maximize its efficiency, the system is equipped with a BR frequency up-conversion mechanism. The BR is connected to an Amplified Piezoelectric Generator (APG) made of a flextensional elastic APX4 steel [46] structure that contains stacked lead zirconate titanate (PZT) ceramics excited in 33-mode (Figure 4). It was originally a APA50XS piezolectric actuator developed by Cedrat Technologies [47], which we use in reverse mode as an electricity generator. A modeling of such a transducer is suggested in [48].

The APG is actuated by two hydraulic cylinders (HC) using the hydraulic energy transmitted from the liquid-filled earplug trough a hydraulic amplifier (HA) and two hydraulic valves (HVs). During a mastication cycle, the TMJ compresses the earcanal wall, which expels the fluid out of the earplug. The HVs lead the fluid alternatively to each HC to actuate the BR mass back and forth. The HA can simultaneously adapt the cylinders’ stroke to the available earplug swept volume and the actuation force of the BR to the comfort pressure of the earcanal (Figure 2).

The BR mass has two equilibrium positions, 
$x_{m}=x_{0}$
 and 
$-x_{0}$
. The EH operates during two major phases. The first is when the BR mass is actuated by the HC, from one equilibrium position toward the unstable position at 
$x_{m}=0$
. During the second phase, the mass reaches and oscillates around the opposite equilibrium position until it stops, while a portion of the vibration energy is converted into electricity by the APG. The mass will move backward when the next mastication action occurs, and this completes one energy harvesting cycle.

A technological solution for the HV is to use a flexible tube buckled by bending. Figure 5 illustrates such an HV integrated on the BR. The HV is made of a flexible tube connected to the hydraulic circuit by rigid mobile and rigid fixed sheaths. The HV is in contact with the mass at point T. The mass motion on the mobile sheath induces a bending angle 
θ
 around rotational point *O* located at the buckled section of the flexible tube. Consequently, a pressure loss is generated through the valve in its buckled position. The HV is to be closed when the BR mass is at a stable equilibrium position (
$x_{m}=\pm x_{0}$
) and to be opened when the mass is at 
$x_{m}=0$
. This sequence of the valve opening and closing will be investigated experimentally in Section 7. Two HVs symmetrically placed on either side of the 
$\vec{z}$
 axis set up an open hydraulic circuit on one side and a closed one on the other, depending on the BR mass’ position. This technological choice for the HV is motivated by three main arguments. First, the absence of dedicated electromechanical transduction compared to the more commonly used actuated valve minimizes the energy losses during the valve’s operation. Second, the post-buckling softening effect of the bent tube minimizes the mechanical impact on the BR mass dynamic operation. Finally, it seems to be well adapted for applications on millimetric scales.

## 4. System Global Modeling

The EH is a multiphysics system composed of hydraulic, mechanical, and electrical components. This section presents the global multiphysics model and describes the system’s theoretical basis of operation. Two coupled subsystems are considered. Section 4.1 shows the modeling of the electromechanical frequency up-converter (BR + APG) under the mechanical influence of the HV and HC. Section 4.2 discusses the design of the hydraulic circuit composed of the two HVs, two HCs, HA, and liquid-filled earplug.

The challenge of this work lies in the experimental validation of the electromechanical converter cycled by two HVs and two HCs. Several hydraulic miniature solutions for HAs and HCs have already been presented in the literature [49,50,51]. The energy loss associated with these components has been neglected in this paper and will be considered in our future works.

### 4.1. Modeling of the Electromechanical Converter

Figure 6 shows the kinematic scheme of the electromechanical converter under the mechanical forces of the HV and HC. The various rigid bodies are defined in Table 1. The BR is shown as four identical arms articulated by eight hinges and a central mass. The hinge’s angle 
φ
 is defined by:
(6)
$$\varphi =\arctan\left(\frac{x_{m}}{L}\right)$$


The APG is considered as a spring of stiffness *K* with an electromechanical coupling 
$k^{2}$
, standing for the piezoelectric transduction and defined by Equation (Equation 7).

(7)
$$k^{2}=\frac{\alpha^{2}}{\alpha^{2}+KC_{p}}$$


The model is based on the following hypotheses:The only mass considered is the BR mass (2).All parts are rigid except for the APG (4).The hinges are considered elastic: They are defined by their rotational stiffness 
$K_{\varphi }$
The mechanical damping of the hinges is included in the global viscous damping coefficient 
$\mu_{air}$
.
$x_{0}\ll L$
.The contact between the BR mass (2) and the HC piston head (5) is considered permanent for 
$0<x<x_{0}$
.

By isolating the BR mass and using Newton’s second law, we can express its dynamic equilibrium. The projection on the 
$\vec{x}$
 axis of the resulting expression is given by Equation (Equation 11). The BR mechanical equilibrium has a mass inertial term, a non-linear stiffness depending on 
$x_{m}$
, the stiffness of the hinge 
$K_{\varphi }$
 and viscous losses. The first term introduced by the HV is related to its rotational stiffness 
$K_{HV}$
 and the second term is related to the friction losses at contact point *T* (see Figure 5). 
$K_{HV}$
 and 
$K_{\varphi }$
 both act in the same way by shifting the stable equilibrium positions of the BR toward 0 as much as they increase. A dry friction model is chosen to describe the dissipation effect of the contact. The rubbing efforts depend on a dry friction coefficient 
$f_{d}$
, related to the nature of the materials that are rubbing together, and by the sign of 
${\dot{x}}_{m}$
. Pressure 
$p_{hc,a}$
 induces the HC hydraulic force, and coefficient 
$\mu_{hc}$
 models the viscous friction at the sealing gasket of the HCs. The hydro-mechanical coupling will be detailed later.

The electrical power generated by the APG is dissipated in a load resistor 
$R_{l}$
, calculated using adaptive impedance matching, as expressed in Equation (Equation 8) [52].

(8)
$$R_{l}=\frac{1}{C_{p}w_{0}}$$


Thus, the electrical equation of the APG is

(9)
$$\frac{U_{p}}{R_{l}}=-\alpha \frac{d}{dt}\left(2\sqrt{{l_{0}}^{2}-x_{m}^{2}}\right)-C_{p}{\dot{U}}_{p}=\frac{2\alpha x_{m}{\dot{x}}_{m}}{\sqrt{L^{2}+x_{m}^{2}}}-C_{p}{\dot{U}}_{p}$$


Consequently, the generated power 
$P_{e}$
 expression is:
(10)
$$P_{e}=\frac{{U_{p}}^{2}}{R_{l}}$$


(11)
$$\begin{array}{cc} m{\ddot{x}}_{m}= & -2K_{eq}\left(\sqrt{x_{m}^{2}+L^{2}}-l_{0})\tan(\varphi \left(x_{m}\right)\right)\;-\mu_{air}{\dot{x}}_{m}-\frac{4K_{\varphi }\varphi }{L}\;-2\alpha U_{p}\tan\left(\varphi \left(x_{m}\right)\right)\\ & \;-\frac{K_{HV}\left(x_{m}\right)\theta \left(x_{m}\right)}{a}(1-f_{d}\mathrm{sign}\left({\dot{x}}_{m}\right))-p_{hc,a}\;S_{hc}-\mu_{c}\;{\dot{x}}_{m} \end{array}$$


Equation (Equation 12) defines the implicit relation between 
θ
, 
$x_{m}$
 and the geometrical parameters of the setup. Additionally, the HV stiffness 
$K_{HV}$
 will depend on its bending angle 
θ
, as shown in Figure 5.

(12)
$$x_{m}=\left[\frac{D_{r}\left(1-\cos\left(\theta \right)\right)+2a\sin\left(\theta \right)}{2\cos(\theta )},\;\;\theta \leq \arctan\left(\frac{2a}{D_{r}}\right)\right]$$


### 4.2. Modeling of the Hydraulic Circuit

In order to model the coupled behavior of the two hydraulic branches, we will use the index “*a*” for the active branch where the fluid flows and the index “*i*” for the inactive one. The following hypotheses will be considered:The flow is incompressible and Newtonian.The hydraulic circuit is rigid (no volume change) and there is no leakage.The hydraulic actuation is considered quasi-static considering the oscillation frequency of the BR.

The mechanical equilibrium is

(13)
$$\mathrm{Inactive}\;\mathrm{cylinder}\;\;\left\{\;p_{hc,i}\;S_{hc}-\mu_{c}\;{\dot{x}}_{hc,i}=0\right.$$


The pressure loss coefficient 
Cf
 for a buckled flexible tube will depend on the HV bending angle 
θ
 and on the BR mass’s position 
$x_{m}$
 (Figure 5). 
Cf
 will be all the more important as 
$x_{m}$
 increases. Its minimal value 
$Cf_{0}$
 is obtained when the HV is unbent. We assume at first that the relationship between 
Cf
 and 
$x_{m}$
 is linear and is defined as follows:
(14)
$$Cf\left(x_{m}\right)=Cf_{0}+\frac{Cf_{c}-Cf_{0}}{x_{0}}\left|x_{m}\right|$$

where 
$Cf_{c}=Cf\left(\pm x_{0}\right)$
. Regarding the BR symmetry, the relation between 
Cf
 and 
$x_{m}$
 must be an even function to enable the system cycling, i.e., 
$Cf\left(x_{m}\right)=Cf\left(-x_{m}\right)$
.

Using Bernoulli’s equation for a current line, we can express the earplug’s internal pressure 
$p_{ear}$
 as a function of the hydraulic resistance in the two parallel branches with Equations (Equation 15) and (Equation 16).

(15)
$$\mathrm{Active}\;\mathrm{branch}\;\;\left\{\;p_{ear}=\frac{1}{a_{h}}\left(p_{hc,a}+Cf_{0}{q_{a}}^{2}\right)\right.$$


(16)
$$\mathrm{Inactive}\;\mathrm{branch}\;\;\left\{\;p_{ear}=\frac{1}{a_{h}}\left(p_{hc,i}+Cf\left(x_{m}\right){q_{i}}^{2}\right)\right.$$

where 
$a_{h}$
 is the hydraulic amplification ratio and *q* is the fluid flow rate.

The no-leakage assumption means that any fluid exciting the earplug flows into the two hydraulic branches (Equation (Equation 17)).

(17)
$$q_{ear}=\dot{\Delta V_{ear}}=q_{a}+q_{i}$$


Moreover, the fluid entering the HC necessarily results in a piston displacement (Equations (Equation 18) and (Equation 19)).

(18)
$$\mathrm{Active}\;\mathrm{cylinder}\;\;\left\{\;q_{a}=S_{hc}{\dot{x}}_{m}\right.$$


(19)
$$\mathrm{Inactive}\;\mathrm{cylinder}\;\;\left\{\;q_{i}=S_{hc}{\dot{x}}_{ic}\right.$$


## 5. Numerical Model and Simulations

### 5.1. Setting the EH Parameters

The energy system combines several variable parameters. To determine these, we first need to identify the fixed technological elements:The transducer is a APA50XS piezoelectric actuator (Cedrat Technologies, Meylan, France) exploited as a generator [47].The hydraulic actuation is ensured by the SMC MQP4-10S HCs, which can operate at 1 kPa pressure [53].The usual size of a hearing aid case is 50 × 20 × 10 mm [54]. The 
L=16
 mm parameter is chosen to be consistent with this scale.

The system was designed by first setting the two coupled requirements to enable the system to work. First, the available swept volume must produce a large enough HC stroke from its equilibrium position until the BR mass reaches 
$x_{m}=0$
 (Equation (Equation 20)). The BR mechanical force on the 
$\vec{x}$
 axis reaches a maximum value 
$F_{c}$
 when 
$x_{m}=x_{c}=\frac{x_{0}}{\sqrt{3}}$
. At this point, a maximum pressure is induced in the earplug and Equation (Equation 21) needs to be verified to remain within the pressure limit. If it exceeds the limit, 
ϵ
 can be decreased, diminishing the extracted energy.

(20)
$$\begin{array}{c} {\left(\Delta V_{ear}\right)}_{max}=\;a_{h}\;S_{hc}\;x_{0} \end{array}$$


(21)
$$\begin{array}{c} p_{c}>\;\frac{1}{a_{h}}\frac{F_{c}}{S_{hc}}\;\;\;\;\mathrm{with}\;\;\;\;F_{c}=\frac{4Kx_{0}\epsilon }{3\sqrt{3}} \end{array}$$


Equations (Equation 20) and (Equation 21) set the amplification level 
$a_{h}$
 and the BR buckling level 
ϵ
 defined in (Equation 22).

(22)
$$\epsilon =\frac{x_{0}}{L}$$


Regarding the technological choices and the system operation requirements, we can evaluate 
n=65%
 (Equation (Equation 5)). This value can be compared to the best theoretical linear energy extraction solution described in Section Equation 4.

### 5.2. Targeted Hydraulic Behavior of the HVs

A multiphysics coupled model has been established with the equations and the preliminary dimensioning previously introduced. The key parameter that remains unknown is the hydraulic behavior 
$Cf(x_{m})$
 of the HVs. The ratio between the pressure loss coefficient of the closed HV and that of the opened HV must be sufficient to force the flow through the opened branch. Thus, we introduced a hydraulic restriction coefficient 
$r_{Cf}$
, which was defined as follows:
(23)
$$r_{Cf}=\frac{Cf(\mathrm{Closed}\;\mathrm{HV})}{Cf(\mathrm{Opened}\;\mathrm{HV})}=\frac{Cf_{c}}{Cf_{0}}$$


The numerical resolution of the coupled differential Equations (Equation 9), (Equation 11), (Equation 13) and (Equation 15)–(Equation 19) can provide the minimum value 
${\left(r_{Cf}\right)}_{min}$
 that needs to be reached to achieve an adequate hydraulic commutation.

### 5.3. Simulation Results

By imposing a volume variation of 
$\Delta V_{ear}\left(t\right)$
 recorded during the mastication cycle of a human subject (Figure 1), the numerical model gives the theoretical temporal evolution of the hydraulic, mechanical, and electrical coupled variables defining the EH. The earplug is assumed to be a perfect flow rate source. The mastication cycle is identically applied four times to test the system’s cycling robustness. The fixed, designed and resulting theoretical parameters of the simulated global model are presented in Table 2a,b.

Figure 7 shows the simulation results of the mass and HCs pistons positions overlaid with 
$\Delta V_{ear}\left(t\right)$
, the flow rates in the two parallel branches and the earplug pressure. The two different sides are identified on the curves by indexes *t* for *“top”* side (
x>0
) and *b* for *“bottom”* side (
x<0
), noting that the effects of gravity are not considered. The same figure shows a focused view on the two main phases of the EH operation during the actuation of the *“bottom”* HC. The simulation begins by the mastication cycle with the following initial conditions:The mass is at the *“bottom”* equilibrium position 
$-x_{0}$
.The *“top”* HV is closed (
$Cf_{top}=Cf_{c}$
) and the *“bottom”* side HV is opened (
$Cf_{bot}=Cf_{0}$
).There is no contact between the mass and the HCs.

During the first phase, the BR mass is pushed until it reaches 
x=0
. The fluid exiting the earplug is mostly guided toward the *“bottom”* HC and pressure is induced in the earplug by the BR counter-reaction force. The mass then crosses 
$x_{m}=0$
 and the second phase begins. The *“top”* HV then opens (
$Cf_{top}=Cf_{0}$
) and the *“bottom”* side HV closes (
$Cf_{bot}=Cf\left(x_{m}\right)>Cf_{0}$
). The fluid is then directed toward the *“top”* side and there is no contact between the mass and an HC. The mass oscillates around 
$x_{m}=x_{0}$
 and energy is harvested by the APG. The adequate hydraulic actuation making the fluid flow toward the right HC is achievable when 
${\left(r_{Cf}\right)}_{min}=26$
.

Figure 8 also presents the different positions of the moving components, the APG voltage, the source and harvested powers and the different energies entering and exiting the system. The theoretical global efficiency of the EH is evaluated at 
$\eta_{g}=85\%$
 with 22 µJ electrical energy harvested from one mastication. 
$\eta_{g}=85$
% is defined as the ratio of the electrical energy 
$E_{e}$
 over the hydraulic energy 
$E_{ear}$
 injected in the system by the earplug. These energies are defined as follows:
(24)
$$E_{e}=\int \frac{U_{p}{\left(t\right)}^{2}}{R_{ch}}\;dt$$


(25)
$$E_{ear}=\int p_{ear}\left(t\right)\;q_{ear}\left(t\right)\;dt$$


The numerical results are derived from the dynamic simulation illustrated in Figure 8. 
$\eta_{g}$
 specifically considers the energy losses in the BR stage and the regular pressure losses in the hydraulic tubing. Given that the latter are theoretically negligible, as mentioned in Section 2.2.2, the energy conversion efficiency of the BR can be approximated as 85%.

The energy losses are mostly located in the BR stage and depend on quality factor *Q*, which has been set in accordance with previous work results on similar BR architectures [52]. The energy source depends on comfort pressure level 
$p_{c}$
 that is set to 1 kPa, i.e., a fraction of the theoretical maximal value of 12 kPa [21]. The EH’s theoretical efficiency could reach a maximum extractable energy of 398 µJ from one mastication cycle (Equation (Equation 5)).

The simulated model shows promising efficiency (Table 2a), and the next section will present the experimental approach to validate the theoretical behavior of the electromechanical converter.

## 6. Experimental Characterizations of the Electromechanical Converter

### 6.1. Description of the Electromechanical Converter

The BR is fabricated by electrical discharge machining (EDM) in a monobloc structure with tolerances under ±10 μm. This method provides better resonator quality factor than when there are assemblies. Figure 9 shows the BR mounted on the experimental test bench. The eight hinges are ensured using four buckled blades (BB), while a vertical guide beam (GB) ensures the APG mounting and prevents any rotations. The local thickenings at the midspan of the blades are intended to facilitate the fabrication. They also maximize the BB compression stiffness, which maximizes the EH efficiency, as we will explain in the following section. The geometrical and mechanical parameters of the BR are referenced in Table 3. The following sections describe the design method that led to the dimensions of the blades, based on the needs of the application.

### 6.2. Designing the BR Beams

In the proposed system, energy harvesting occurs during the oscillation phase. The main issue is to maximize the energy conversion efficiency 
$\eta_{br}$
 of this stage. Richards et al. demonstrated the piezoelectric transduction at the resonance frequency can be approximated with Equation (Equation 26) [33]. This expression is valid for an electric extraction based on the adaptive impedance-matching strategy.

(26)
$$\eta_{br}=\frac{k_{sys}^{2}\;Q}{k_{sys}^{2}\;Q+2}$$

where 
$k_{sys}^{2}$
 is the global electromechanical coupling coefficient of the harvesting system. The latter can be expressed with the electromechanical parameters of the APG and the stiffness of the beams (Table 3) as follows:
(27)
$$k_{sys}^{2}=\frac{\alpha_{eq}^{2}}{\alpha_{eq}^{2}+\left(K+K_{gb}\right)C_{p}}$$

where 
$\alpha_{eq}$
 is the equivalent Piezoelectric Force Factor (PFF) given by Equation (Equation 28). It depends on the APG PFF and on the ratio between the equivalent stiffness 
$K_{eq}$
 (Equation (Equation 29)) of the structure on the 
$\vec{z}$
 axis and the equivalent stiffness of the APG mounted on the GB.

(28)
$$\alpha_{eq}=\alpha \;\frac{K_{eq}}{K+K_{gb}}$$


(29)
$$K_{eq}=\frac{K_{bb}\left(K+K_{gb}\right)}{K_{bb}+K_{gb}+K}$$


Equations (Equation 26)–(Equation 29) combined demonstrate that in order to maximize 
$\eta_{br}$
 we must make sure that 
$K_{bb}\gg K\gg K_{gb}$
. If the condition is respected then 
$K_{eq}$
 can be assimilated to *K* and 
$\alpha_{eq}\approx \alpha$
.

When the mass crosses the unstable 
$x_{m}=0$
 position, the compression force 
$F_{z,0}$
 induces the maximum stress on the BB. 
$F_{z,0}$
 can be expressed by Equation (Equation 30). 
$K_{bb}$
 can then be assured if the compression efforts absorbed by the BBs do not induce any secondary buckling mode during the BR operation.

(30)
$$F_{z,0}=2K\left(\sqrt{x_{0}^{2}+L^{2}}-L\right)$$


The beams design was obtained using a FEM model whose results are summarized in Figure 10. It reveals that the first secondary undesirable buckling mode appears for 
$F_{z,0}=6.4$
 N. If this happens, the compression force is not transmitted from the mass to the APG, 
$K_{bb}$
 is not assured and 
$k_{sys}^{2}$
 is diminished (Equation (Equation 27)). Thus, we establish a buckling height domain (Figure 10) to avoid the secondary buckling for the fabricated BR. The preliminary simulated case presented in the previous section is in the preferable domain. Moreover, the force absorbed by the APG must be under the maximal limit of 18 N above which the device could become damaged.

### 6.3. Experimental Characterizations

The experimental test bench for the characterizations of the electromechanical converter is presented in Figure 9. An additional beam is mounted with the APG to prevent any rotation whatsoever. A micrometric screw sets the initial buckling level to 
$x_{0}$
 (
±10
 µm). The bearing decouples the rotation of the screw with the beam structure to prevent unwanted torsional stress. The central mass at 45° receives and reflects the displacement sensor-emitted laser along the 
$\vec{z}$
 axis and captures the 
$x_{m}$
 position. The APG power is dissipated in the 
$R_{l}$
 resistor and the 
$U_{p}$
 voltage (
±1
 mV), along with 
$x_{m}$
, are monitored through a data acquisition card NI-USB 6212 and a LabVIEW interface. Finally, the two HC plungers actuate the mass from one stable equilibrium position to the other. The HV mechanical influence is not considered here in order to focus on the electromechanical converter dynamic behavior.

The mass is pushed by a HC from a stable equilibrium position until it reaches 
$x_{m}=0$
 and then oscillates around the opposite stable equilibrium position. The experimental data have been fitted on the oscillation phase of the simulated global model, as the test bench does not reproduce the hydraulic circuit behavior yet. These data were used to identify and recalibrate the following electromechanical parameters of the EH through iterative simulations of the global model: *Q*, 
$f_{0}$
, *K* and 
$k_{sys}^{2}$
 (Table 4). The experimental oscillation frequency 
$f_{0}$
 and energy conversion efficiency 
$\eta_{br}$
 were also calculated.

Experimental 
$K_{bb}\approx K/2$
 can be calculated from the identified 
$K_{eq}$
 with Equation (Equation 29), supposing 
$K_{gb}$
 is negligible (Table 3).

The vibration of the BR does not produce any noise experimentally, but the integrated system’s acoustic influence has not been modeled. Nevertheless, the 50 Hz vibration frequency should not disturb the hearing, as it should be packaged in such a way that the BR frame is isolated from the skull. Moreover, recent studies show that the conventional bone vibrators can deliver a stimuli signal under 400 Hz, even when in direct contact with the cranial bones [55].

### 6.4. Analysis and Summary

The experimental dynamic behavior of the converter was predictable using the equations established once the system parameters, listed in Table 4 had been identified. However, the performances of the experimental prototype are below the theoretical expectations (Table 2a). This efficiency discrepancy is mainly associated with the quality factor and the electromechanical coupling coefficient drop (Equation (Equation 26)).

The decrease in quality could be related to the general non-permanent assembly that uses non-perfect embeddings with fixation screws and a mobile micrometric screw. Small undulations were observed on the BBs after fabrication and may also have gotten worse during the manipulation. These undulations increase the probability of secondary buckling, which contributes to reducing 
$K_{bb}$
 and diminishing 
$k_{sys}^{2}$
, according to Equations (Equation 27)–(Equation 29). Improvements are proposed further on to resolve this issue and reduce the energy losses during this stage.

## 7. Experimental Approach of the HV Design

To validate the feasibility of the HVs, we adopted an experimental approach to find the material and geometrical parameters of the tube to achieve 
${\left(r_{Cf}\right)}_{min}=26$
 (Table 2b). A buckled kapton tube [56] of internal diameter 
$D_{t}=1.05$
 mm and thickness 
$th_{t}=25$
 µm was found to meet this hydraulic criteria. It will be referred to as HV_*T*1_.

### HV Hydraulic Experimental Characterization

Figure 11 shows the HV_*T*1_ during the experimental characterizations on the hydraulic test bench. An electromechanical rotary plate is monitored on the LabVIEW interface and imposes a bending angle 
θ
. A liquid-filled syringe sets the fluid flow through the HV_*T*1_ and 
$q_{syr}$
 is evaluated by tracking the syringe piston displacement. The pressure loss coefficient 
$Cf_{T1}$
 is calculated as follows:
(31)
$$Cf_{T1}=\frac{p_{1}-p_{2}}{q_{syr}^{2}}$$

where 
$p_{1}$
 and 
$p_{2}$
 are the fluid pressures monitored before entering and after exiting the HV_*T*1_, respectively.

The tests were repeated seven times for 
θ
 varying from 
$0^{\circ }$
 to 
$60^{\circ }$
 with 
$10^{\circ }$
 increments. The experimental results are plotted in Figure 12.

The hydraulic restriction coefficient of the HV_*T*1_ has been evaluated to 
$r_{Cf}60$
 if we consider the opened angle 
$\theta_{0}=20^{\circ }$
 and the closed angle 
$\theta_{c}=60^{\circ }$
. Its hydraulic behavior respects the minimum criteria of 
${\left(r_{Cf}\right)}_{min}=26$
, and therefore can ensure the adequate hydraulic commutation to cycle the BR mass motion. In order to set an optimal angle operation range 
Δθ
, we must investigate the evolution 
$K_{HV}\left(\theta \right)$
. This will not be discussed in this paper. However, 
Δθ
 must be minimized in order to limit the energy needed for the HV’s operation and also, 
$\theta_{0}$
 must be minimized to reduce the hydraulic energy losses in the active branch.

The following section presents the EH model using the hydraulic characterization data of the HV_*T*1_ and the electromechanical experimental data of the BR with the APG.

## 8. Global Model Recalibration with Experimental Data

The experimentally identified parameters have been implemented in the global multiphysics coupled model to enhance its predictability and to evaluate the EH performances. The mechanical influence of the HV_*T*1_ is identified with an experimental approach, characterizing the energy losses at the HV-mass contact point (T on Figure 5).

### 8.1. Recalibration with Release Trials

The HV_*T*1_ was mounted onto the experimental test bench presented in Figure 9, and two experimental trials were performed to identify the HV mechanical influence: free oscillations and oscillations with the presence of the HV_*T*1_, during a harvesting phase. Thus, the difference in the dynamics between the two tests focuses on the HV_*T*1_’s mechanical influence.

The mass is initially set to 
$x_{m}=x_{0}$
 and then pushed toward 
$x_{0}$
. The HV_*T*1_’s internal pressure is adjusted to 30 kPa with an oil column, in order to be consistent with the simulated case (Table 2b). The oscillation phase begins when the mass crosses the 
$x_{m}=0$
 position. In order to set 
$\theta_{0}$
, we added two angle stoppers, illustrated in Figure 13. 
$\theta_{c}$
 was then reached when the oscillations stopped at 
$x_{m}=x_{0}$
. The mass position and the APG tension were monitored during the tests.

The global model parameters listed in Table 5 have been recalibrated for both test configurations by iterative simulations. The simulations with the recalibrated parameters and the experimental data are superimposed in Figure 14 concerning the harvesting phase of both configurations.

After re-calibration, the good match between the simulated and the experimental data verifies that the electromechanical dynamic behavior of the EH is theoretically predictable by the established model. The HV_*T*1_ stiffness shifts 
$x_{0}$
 toward 0 and its value at 
$\theta_{c}$
 is evaluated with the shift gap and Equation (Equation 11). The HV_*T*1_ adds a supplementary dynamic mass in the BR and changes the oscillation frequency. The electromechanical coupling coefficient is not influenced by the presence of the HV. The quality factor decreases and the dry friction coefficient is calculated from the drop. Moreover, *a* and 
Δθ
 are graphically evaluated.

### 8.2. Analysis and Discussion

The experimental hydraulic behavior of the HV_*T*1_ is theoretically capable of providing 
$r_{Cf}\geq 26$
 and ensuring an adequate hydraulic commutation cycling for the BR mass’s motion.

The quality factor decrease is due to the rubbing between the mass and the HV and diminishes the BR energy conversion efficiency. The friction losses can be minimized by an order of magnitude by using materials with a low friction coefficient, like Teflon [57]. The HV-mass contact can also profit from magnetic interactions instead of direct rubbing.

The global electromechanical coupling is not influenced by the presence of the HV, which is consistent with Equation (Equation 27). Most of the energy loss is due to the 
$k_{sys}^{2}$
 drop as a result of the softening of the BBs from the prototype fabrication and manipulation. It can be minimized by using thicker BBs while taking care that the buckling domain does not reach the critical point where the APG could be damaged (Figure 10). With BBs having a thickness of 200 µm instead of 75 µm, we can expect 26 % improvement in 
$\eta_{br}$
 (Equations (Equation 26)–(Equation 29)).

### 8.3. Directions for Improvements

The EH is defined by the multiphysical complex coupling of numerous parameters, calculated from two key pieces of information characterizing the source: the comfort pressure 
$p_{c}$
 and the displacement 
${\left(\Delta V_{ear}\right)}_{max}$
 of the earplug. For design purposes, the flow rate is assumed to be known and imposed (Figure 1). It is then possible, with the results obtained at the output of the numerical model, to establish the system evolution trend depending on the variation of these parameters. Table 6 provides a summary of the influence of each parameter on the system, based on the maximization of two criteria: the global energy conversion efficiency 
$\eta_{g}$
 and operation 
$\tilde{f}$
. 
$\eta_{g}$
 refers to the energy conversion efficiency of the overall system for one chewing cycle. 
$\tilde{f}$
, on the other hand, refers to the system’s ability to meet the operating specifications for all components with a given set of parameters. If this criterion is not respected, the system does not function; consequently, the system efficiency is equal to zero. The 
$\tilde{f}$
 criterion of the EH is then defined as follows:

$(r_{Cf})\geq 26$
The resonator stays bistable with the influence of 
$K_{HV}$
.No mass-piston collision during the oscillation phase.Equations (Equation 20) and (Equation 21) are satisfied.

Table 6 shows the qualitative relationship between a parameter and a criterion. A positive relationship (Po) means that the parameter has to be maximized in order to positively influence the concerned criterion. In contrast, a negative relationship (Ne) dictates that it should be minimized to positively influence the concerned criterion. The complexity of the coupling between the different parameters is reflected in the inverse proportionality of their influence on the efficiency and operating criteria. Indeed, a simple modification that theoretically improves the system’s efficiency can lead to a critical malfunction. Several compromises must then be found to maximize the efficiency and guarantee the system operation. A next step would be to draw up an experimental design with these parameters in order to develop an algorithm capable of optimizing the system settings for a given energy input (
$p_{ear}$
, 
$\Delta V_{ear}$
).

### 8.4. Power Capability of the Proposed Energy Harvester

With the analyses that led to Table 6, it is possible to approach the optimal set of parameters with iterative simulations for a given individual (Table 7). By maximizing the 
$\eta_{g}$
 with the constraints related to 
$\tilde{f}$
, we simulated the energy harvester response for two different configurations: the ideal one and the actual one. The first configuration supposes that the 
$k_{sys}^{2}$
 drop was compensated by thickening the BBs and the friction loss coefficient between the mass and that the HV has been improved by an order of magnitude by using a Teflon/Teflon interface [57]. The second configuration involves all the defaults of the components that have been experimentally characterized to date. Lastly, the hydraulic and mechanical properties of the HV_*T*1_ were used for both simulations with 
$r_{Cf}=26$
. Using a manually iteratively optimized design that takes into account the aforementioned fabrication defaults, we can anticipate an average power output of 13 μW for a mastication cycle. Furthermore, by implementing straightforward experimental corrections, we can anticipate harvesting 359 μW from the same energy source, thereby providing a promising incentive for further exploration in this area.

This type of EH architecture has never been implemented for cochlear implants. Nevertheless, Table 8 compares different designs of bistable piezoelectric EH with our work and Table 9 situates the proposal within different types of EHs in the human earcanal environment. The electromechanical coupling of piezoelectric energy harvesters relies heavily on the directional excitation operation mode. While many operate in the 
$d_{31}$
 mode under bistable configurations, for improved integration, it is important to note that transduction in the 
$d_{31}$
 mode is less efficient compared to what can be achieved with the 
$d_{33}$
 mode [58].

## 9. Future Work

### 9.1. Integration Issues

This work aims to demonstrate the theoretical and experimental feasibility of a new EH architecture for the exploitation of the earcanal wall deformation as an energy source. The energy extraction with a hydraulic solution allows the use of the available space inside a conventional hearing aid casing fitting behind the ear. However, the integration aspects will be addressed in further work. The following lines list the critical points that are in need of attention.

#### 9.1.1. Bio-Compatibility

Superior functionalities are necessary for biomedical devices compared to conventional ones. These include high flexibility and stretchability, seamless integration with the human body, exceptional adaptability, robust bonding with rigid electronics, and long-term operation [62]. 

Figure 3 showcases the inflatable earplug for energy-harvesting purposes. This earplug is a sealed, medical-grade certified product developed by Sonomax [20]. The flexibility and adaptability criteria are easily met through the water–inflated silicone interface.

Additionally, research has demonstrated the safe encapsulation of PZT ceramics for implantable and epidermal devices [63,64,65,66]. Furthermore, medical-grade flexible tubes, commonly used in hospitals, readily meet the hydraulic circuit requirements. With these factors in mind, it is important to note that the device, earplug aside, should be housed within a plastic casing, facilitating the overall prototype’s bio-compatibility.

#### 9.1.2. Power Management and Storage

An energy extraction electronic circuit has to be implemented with the energy harvester. A classic maximal power point tracker (MPPT) circuit could be used [67]. Zhang et al. conducted a comprehensive analysis comparing various charging circuits specifically designed for piezoelectric energy harvesters [68]. Their findings indicate that the optimal charging solution for each application primarily relies on factors such as the electromechanical coupling of the system, the excitation frequency, and the storage capacitance. However, a quantitative analysis to determine the most suitable solution for our specific application has not yet been performed.

#### 9.1.3. Geometrical Design

The BR’s overall length has to be adapted to the available space, depending on the device’s casing size. The BR buckling level has to be readjusted in consequence. The model presented here, and the results summarized in Table 6, should be helpful for redesigning a functional and efficient energy harvester for any individual.

Moreover, the HVs have to be positioned around the BR to ensure their buckling through the dynamic mass movement.

### 9.2. Other Applications

The multiphysics coupled model introduced above, as well as the analysis of the system summarized in Table 6, can lead to the development of an integrated EH for earcanal wall-deformation exploitation. Nevertheless, The earcanal wall deformation could be considered as a worst-case scenario for such an EH. In fact, the restricted volume of the hearing aid casing, the low power density of the energy source, and the soft tissue’s interface imply numerous engineering challenges. Noting that these can be settled in future work, other applications could benefit from the integration of the harvesting system presented here. Indeed, the system’s strength lies in its ability to transform a one-way “force” into a two-way “force”. This gives access to interesting energy sources for the autonomy of portable devices. For example, the order of magnitude of the pressure applied by the finger on a push-button is about 60 kPa. Considering a reasonable volume of 1 mL of displaced fluid during the pushing phase, the EH could exploit 40 mJ from one push if actuated with the finger. With 10% of efficiency, the 4 mJ harvested energy could be sufficient enough to supply Zigbee-based [69,70] or Bluetooth low-energy-based [71] transmitters integrated in smart wireless devices, such as light switches or other typical applications requiring a push-button.

Finally, this work provides a step forward to future in-ear hybrid energy harvesting for the complete autonomy of in-ear devices. Indeed, hybrid energy harvesters can exploit multiple energy sources at the same time [68,72] and thus, are currently the most promising solution for autonomous wearables. The EH concept presented here exploits the local compression energy, and it can be coupled with flexible piezoelectric materials integrated in the earplug in order to harvest the energy due to the flexion of the earcanal wall during mastication as well [15]. Again, several studies show viable thermal energy harvesters profiting from the body heat with thermoelectric generators [73] that can also be integrated around the earplug. Nevertheless, more progress must be made in the development of power management systems’ amplitude, frequency, transduction processes, etc., in order to integrate multiple transducers for hybrid energy harvesting [11].

## 10. Conclusions

This work emphasizes the need to enhance the autonomy of the batteries of devices such as hearing aids and the exploitation of the mechanical deformation of the earcanal as a solution.

A new EH architecture maximizing the energy conversion efficiency is proposed. A hydraulic, mechanical and electrical coupled model is established. Its numerical evaluation predicts that the system can theoretically harvest 85% of the available energy with a hydraulic extraction method. The dynamic behavior and the performances of the fabricated electromechanical converter composed of a BR and APG are evaluated. A new concept of HVs is proposed based on the buckling of flexible tubes. The proposed HV design is further investigated experimentally to verify if the required theoretical behavior extracted from the numerical simulations is achieved.

The experimental data were used to recalibrate the EH multiphysics model and to evaluate its performances. The global predicted efficiency is 
$\eta_{g}=2.6\%$
. The energy losses factors are identified and solutions are proposed to increase the efficiency to 26%.

## Figures and Tables

**Figure 1 micromachines-15-00415-f001:**
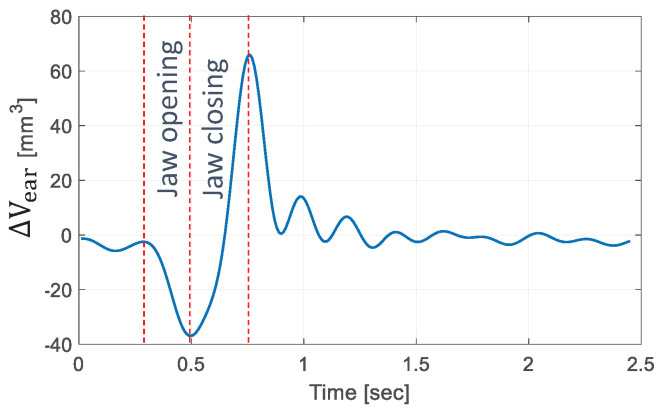
Earcanal volume variation for one mastication cycle.

**Figure 2 micromachines-15-00415-f002:**
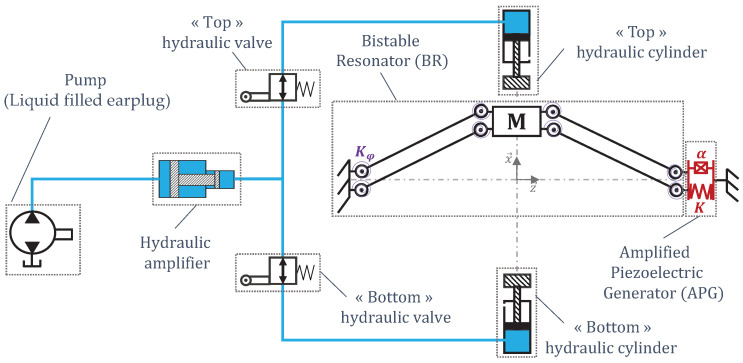
Schematic presentation of the frequency-up converted piezoelectric EH exploiting the earcanal’s geometric variation.

**Figure 3 micromachines-15-00415-f003:**
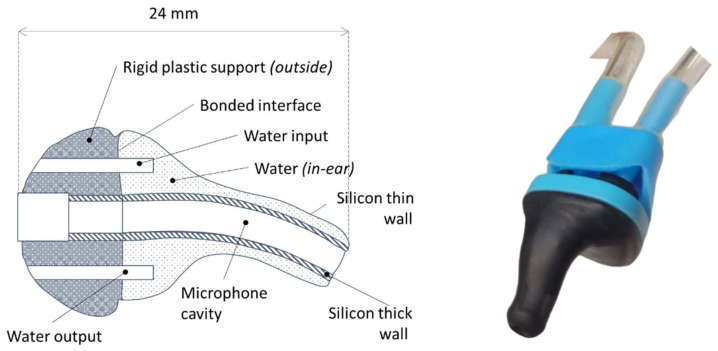
Earplug presentation. (**a**) Earplug schema. (**b**) Earplug picture.

**Figure 4 micromachines-15-00415-f004:**
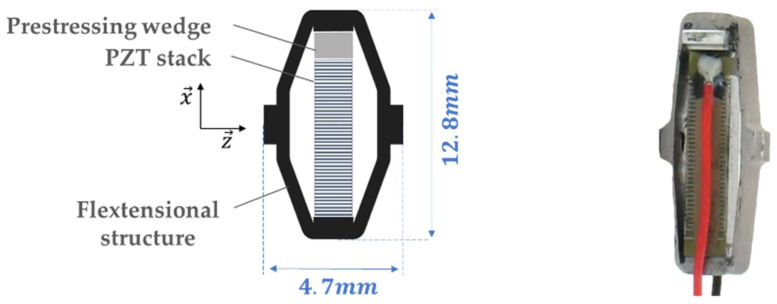
Amplified piezoelectric generator (APG). (**a**) APG detailed schema. (**b**) APG picture.

**Figure 5 micromachines-15-00415-f005:**
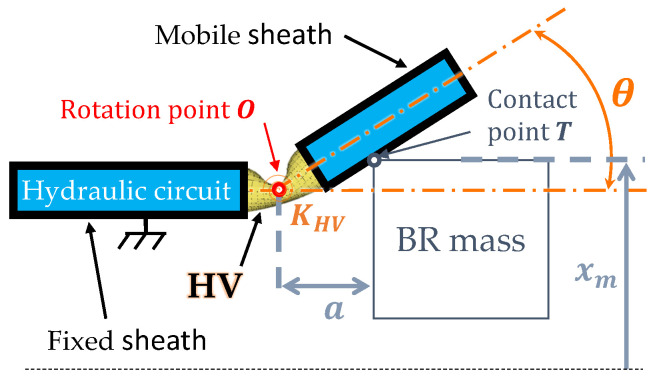
Details of the contact between an HV and the BR mass.

**Figure 6 micromachines-15-00415-f006:**
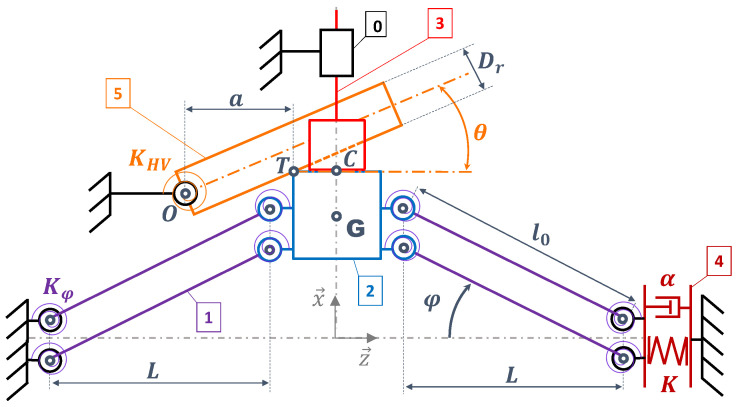
Kinematic scheme of the electromechanical converter under the mechanical influence of a HV and HC.

**Figure 7 micromachines-15-00415-f007:**
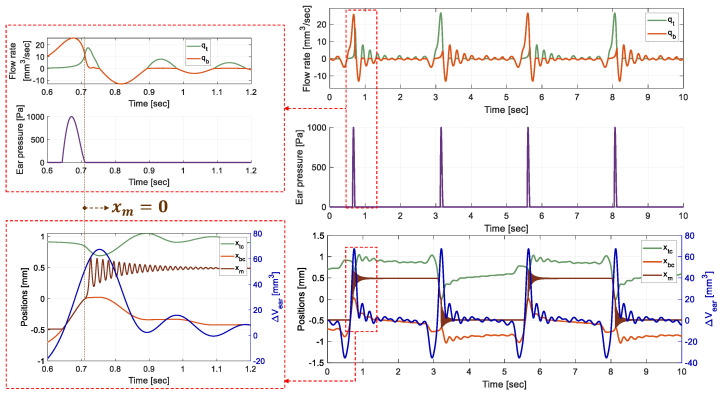
Positions of the mass and HCs’s pistons, overlaid with 
$\Delta V_{ear}\left(t\right)$
, the flow rates in the two parallel branches and the earplug pressure.

**Figure 8 micromachines-15-00415-f008:**
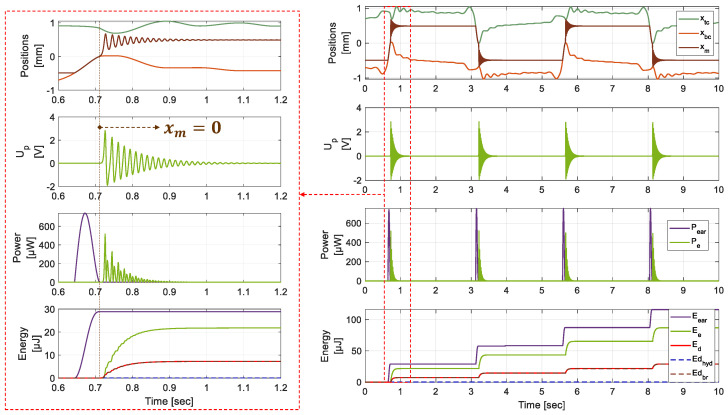
Positions of the mass and HCs’s pistons, APG voltage, harvested and source powers and different energies entering and exiting the system.

**Figure 9 micromachines-15-00415-f009:**
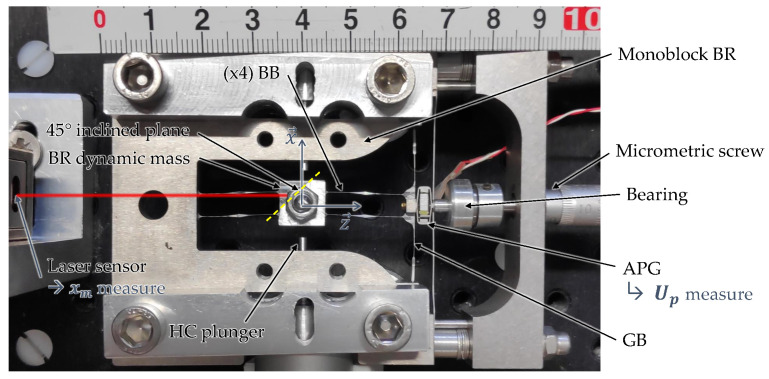
Test bench of the electromechanical converter (metric units).

**Figure 10 micromachines-15-00415-f010:**
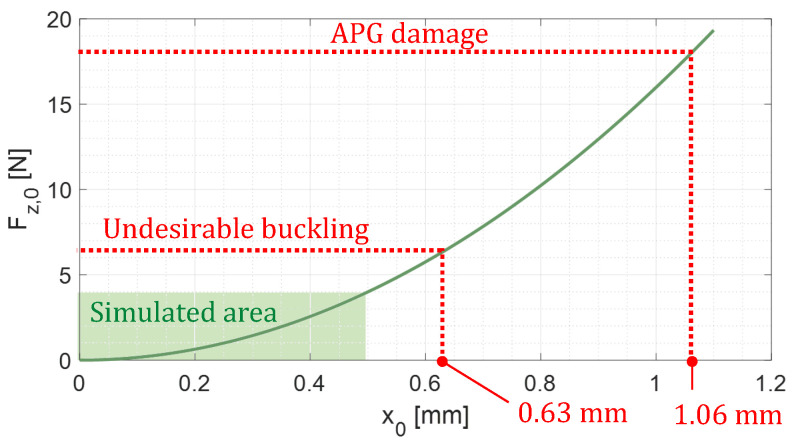
APG elastic counter reaction on 
$\vec{z}$
 axis at 
$x_{m}=0$
, depending on the initial structural buckling height 
$x_{0}$
.

**Figure 11 micromachines-15-00415-f011:**
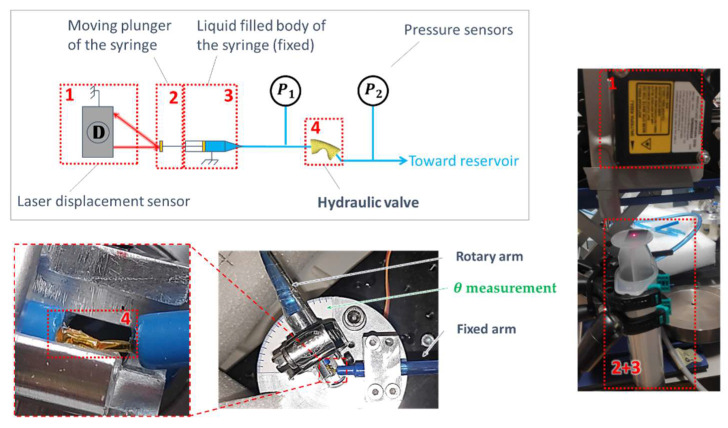
Schematic view and pictures of the hydraulic characterization test bench for the HVs.

**Figure 12 micromachines-15-00415-f012:**
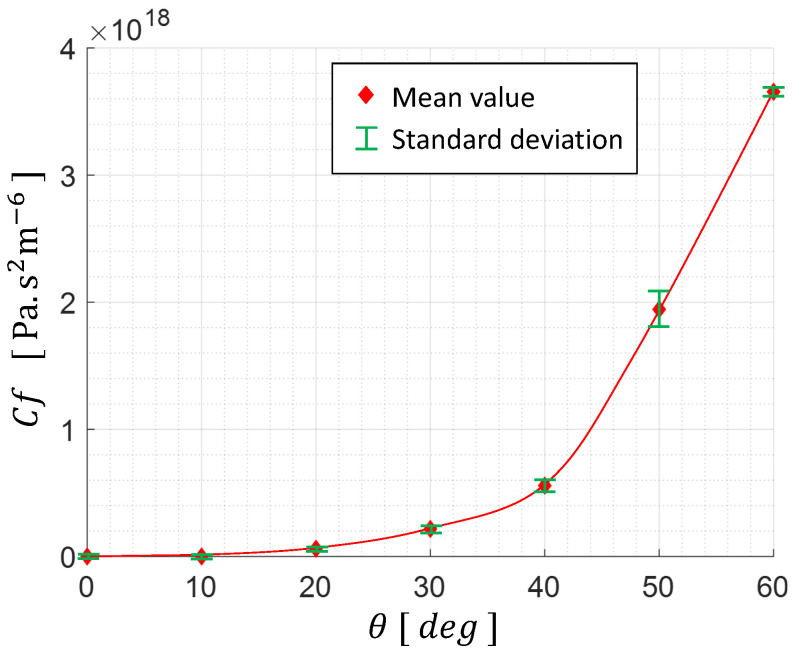
Hydraulic characterization of the HV_*T*1_.

**Figure 13 micromachines-15-00415-f013:**
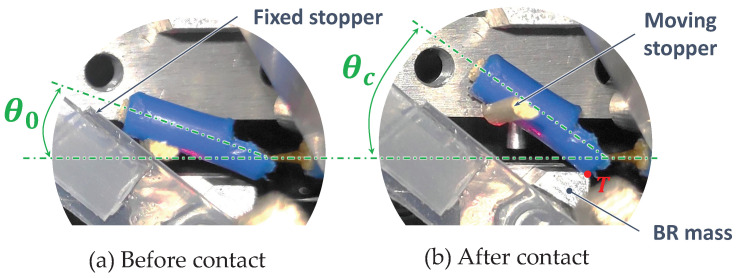
HV_*T*1_ picture for (**a**) 
$\theta_{0}\left(x_{m}=-x_{0}\right)$
 and for (**b**) 
$\theta_{f}(x_{m}=x_{0}$
).

**Figure 14 micromachines-15-00415-f014:**
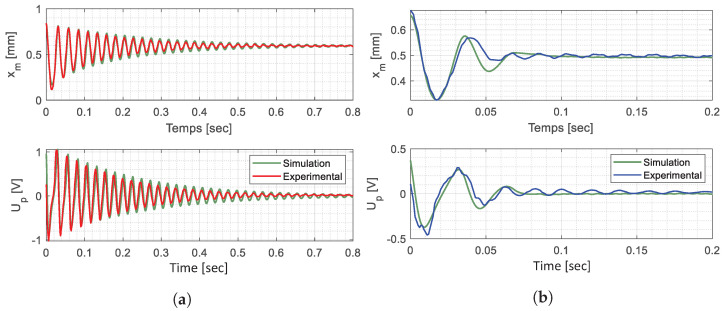
Experimental measurements and global system simulation with the identified parameters of Table 5. (**a**) Free oscillations. (**b**) Oscillations with HV_*T*1_ presence.

**Table 1 micromachines-15-00415-t001:** Definition of Figure 6 bodies.

Num	Name
0	Fixed frame
1	BR arm
2	BR mass
3	HC piston head
4	APG
5	Simplified HV mechanical model

**Table 2 micromachines-15-00415-t002:** Simulated model theoretical parameters.

Parameter	Value
(a) Electromechanical	
$K_{eq}$ [N/µm]	0.256
$x_{0}$ [mm]	0.49
*L* [mm]	16.0
*m* [g]	5.88
$x_{c0}$ [mm]	0.69
$R_{l}$ [kΩ]	6.39
α [N/V]	0.105
$C_{p}$ [µF]	0.25
*Q* [-]	50
$\eta_{g}$ [%]	85
(b) Hydraulic	
$D_{hc}$ [mm]	4
$a_{h}$ [-]	29
$p_{c}$ [kPa]	1
${\left(\Delta Vear\right)}_{max}$ [mm^3^]	60
$Cf_{0}$ [Pa·s^2^/m^6^]	0.21 × 10^17^
$r_{Cf}$ [-]	26

**Table 3 micromachines-15-00415-t003:** Definitions and values of the fabricated BR.

Parameter Definition	Symbol	Value [Unit]
APX4 steel Young’s modulus	*E*	211 [GPa]
APX4 steel elastic resistance	$Re_{APX4}$	955 [MPa]
Dimensions of a buckled beam	*L* × *l* × *e*	16 × 0.07 × 1.2 [mm]
Dimensions of the guide beam	$L_{v}$ × $l_{v}$ × *e*	17.5 × 0.07 × 1.2 [mm]
Soft hinge stiffness	$K_{\varphi }$	0.006 [N/m/rad]
Stiffness of the guide beam along $\vec{y}$	$K_{gb}$	190 [N/m]
Stiffness of four buckled blades along $\vec{z}$	$K_{bb}$	1402 [kN/m]
Stiffness of the APG along $\vec{z}$	*K*	252 [kN/m]

**Table 4 micromachines-15-00415-t004:** Theoretical and experimentally recalibrated values of the electromechanical converter’s parameters.

Symbol	Simulation with Theoretical Parameters	Simulation with Recalibrated Parameters
*Q*	50.0	30.0
$f_{0}$	47.0 Hz	27.9 Hz
$x_{0}$	0.49 mm	0.50 mm
$K_{eq}$	2.56 × 10^5^ N/m	0.85 × 10^5^ N/m
$K_{bb}$	14.2 × 10^5^ N/m	1.27 × 10^5^ N/m
$k_{sys}^{2}$	16%	1.25%
$\eta_{br}$	85%	12.9%

**Table 5 micromachines-15-00415-t005:** Experimentally recalibrated values of the EH parameters with and without the presence of the HV_*T*1_.

Parameter	BR	BR + HV_*T*1_
$D_{r}$ [mm]	–	4
Δθ [deg]	–	[≈19; ≈36]
$K_{T1}\left(\theta_{c}\right)$ [Nmm/rad]	–	0.27
*a* [mm]	–	2.24
$f_{d}$ [-]	–	0.42
*K* [N/m]	84,480	84,480
$x_{0}$ [mm]	0.59	0.50
*Q* [-]	24.0	5.0
$k_{sys}^{2}$ [%]	1.25	1.25
$\eta_{br}$ [%]	12.9	2.6
$R_{l}$ [k]	15.5	15.5
*m* [g]	5.88	9.00
$f_{0}$ [Hz]	32.9	27.9

**Table 6 micromachines-15-00415-t006:** Summary of key parameters’ influence on the system operation capability and its overall efficiency.

Symbol	Influence on the System	Proportionality
ηg ^(*)^	f˜ ^(**)^
*a*	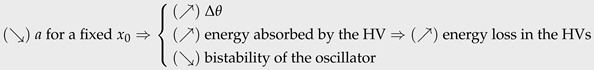	Po ^(1)^	Ne ^(2)^
θ0	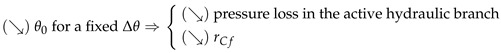	Ne	Po
*m*	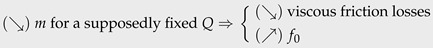	Ne	Ne
tht	(↘)tht⇒(↘)KVH⇒(↘)energyabsorbedbytheHV⇒(↘)energylossintheHV	Ne	Ne
Dt	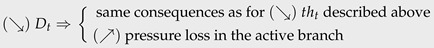	Ne	Ne
Dhc	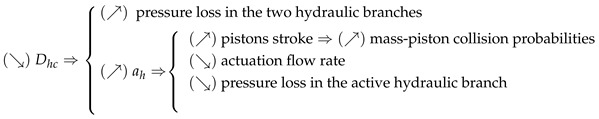	Ne/Po	Ne/Po
x0	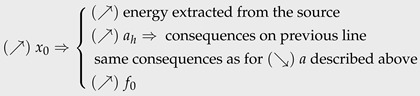	Po	Ne/Po
*L*	(↘)L⇒ same consequences as for (↗)x0 described above	Po	Ne/Po
*K*	(↗)K⇒ same consequences as for (↗)x0 described above	Po	Ne/Po
α	(↗)α⇒(↗)ksys2	Po	Po

^(1)^ Positive proportional influence. ^(2)^ Negative proportional influence. ^(*)^ Efficiency criteria. ^(*)^ Operation criteria. 
(↘)
 A decrease. 
(↗)
 An increase.

**Table 7 micromachines-15-00415-t007:** Simulation results of the manually optimized parameters of the EH designed for the earcanal volume variation shown in Figure 1 and 
$p_{c}=12$
 kPa (for one mastication cycle).

Parameter	Ideal Experimental Design	Actual Experimental Design
*L* [mm]	10	
$D_{hc}$ [mm]	4	
$p_{c}$ [kPa]	12	
${\left(\Delta V_{ear}\right)}_{max}$ [mm^3^]	60	
$x_{0}$ [mm]	0.91	0.89
*m* [g]	3.0	3.0
$R_{l}$ [kΩ]	3.66	3.74
$a_{h}$ [-]	3.8	3.5
$f_{0}$ [Hz]	189	185
$\max(E_{hyd})$ [µJ]	720	
$E_{ear}$ [µJ]	362	340
$P_{e}$ [µW]	359	13.3

**Table 8 micromachines-15-00415-t008:** Evaluation of the proposal in comparison to different designs of piezoelectric bistable EHs reported in the literature.

Excitation Mode	Output Power [µW] ^(*)^	LFA Freq. [Hz]	HFT Freq. [Hz]	Specific Surface [mm^2^]	Ref.
$d_{31}$	2.4	20.8	263	60 × 10	[43]
$d_{31}$	143	4–6	NA	100 × 100	[36]
$d_{31}$	1000	10.7–20	NA	150 × 149	[59]
$d_{31}$	3.4	100–200	NA	2 × 2 (MEMS)	[29]
$d_{33}$	1.7 ^(1)^|66 ^(2)^	1.5	60	70 × 70	This work

^(1)^ Mean power for a mastication cycle (1.5 Hz). ^(2)^ Mean power during BR oscillation (60 Hz). ^(*)^ Experimental data.

**Table 9 micromachines-15-00415-t009:** Evaluation of the proposal in comparison to different types of EHs suitable for applications in the human earcanal environment.

Energy Source	Transduction Method	Application	Output Power [µW] ^(*)^	Power Density [mW/cm^3^]	Source Freq. [Hz]	Ref.
Tympan vibration	Piezoelectric (PLD-PZT)	Cochlear implant	16.3 ^(*p*)^	1.5 ^(*p*)^	1780	[3]
Inner ear	Biofuel cell	Wireless sensors	10^−3^	NA	NA	[60]
earcanal wall flexion	Piezoelectric (PVDF)	Cochlear implant	70	0.04	1.5	[15]
Skin deformation	Piezoelectric (PVDF-PTrFE)	Wearable electronics	10^−3^	0.5	NA	[61]
earcanal wall compression	Piezoelectric (PZT)	Cochlear implant	1.7 ^(1)^|66 ^(2)^	NA	1.5	This work

^(1)^ Mean power for a mastication cycle (1.5 Hz). ^(2)^ Mean power during BR oscillation (60 Hz). ^(*)^ Experimental data. ^(*p*)^ Peak value.

## Data Availability

The raw data supporting the conclusions of this article will be made available by the authors on request.

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
