# Peer review of "A Novel Piezoelectric Energy Harvester for Earcanal Dynamic Motion Exploitation Using a Bistable Resonator Cycled by Coupled Hydraulic Valves Made of Collapsed Flexible Tubes"

_micromachines, 2024, doi:10.3390/mi15030415_

Round 1

Reviewer 1 Report

Comments and Suggestions for Authors

The manuscript discusses the topic of micro energy harvesting, which is important for hearing aids. However, there are some issues that are unclear and need to be addressed before the manuscript can be accepted.

(1)   Dynamics optimization or bistable design of micro piezoelectric energy harvesters has been a well-worn topic. Authors need to review the results of the literature in this field and summarize their own new contributions.

Ref:

Nano Energy, 2022, 102: 107602.

Energy, 2021, 228: 120595.

IEEE Sensors Journal, 2023, 23(4): 3521-3531.

(2)   Are the hydraulic and electromechanical models coupled or decoupled? Which of them is involved in the optimized design of this manuscript? The authors need to clearly state it and decrease the length of irrelevant content.

(3)   What are the objective function and constraints of the optimization problem?

(4)   What role does bistability play in this design? Can it be analyzed and discussed quantitatively through the potential energy function or other tools?

(5)   Lack of performance comparison with literature.

(6)   Lack of quantitative analysis on energy supply and demand. Does this improvement in energy harvester performance contribute sufficiently to hearing aid life extension?

Author Response

Dear reviewer#1,

Please find the point by point responses to your questions in the attached document.

Best regards,

Dr. Avetissian

Reviewer 2 Report

Comments and Suggestions for Authors

The manuscript reports a new energy harvester concept for a liquid-filled earplug transferring energy outside the earcanal to a generator. Compared to previous literature, the authors suggested an unique piezoelectric energy harvester. Overall, this manuscript is systematically organized with sufficient supplement materials. Also this include some originality. For publication, however, the authors need to address the following points.

-, This model shows a theoretical energy conversion efficiency of 85%. What is a reference? How to calculate the theoretical efficiency?

-, A reviewer can't image the implantation feature for all-in-one system. The authors need to provide more details. For example, an energy storage device such as batteries and supercapacitor require DC to charge it. This necessitates the use of a DC power source, as AC power would require rectification. Even if the authors described ‘a low energy-conversion efficiency’, a key issue is ‘low-freqency’ energy. Moreover, wearable devices that monitor biological signals may encounter material restrictions, which the authors should address in future research. Please see the following articles as reference [EcoMat 4 (2022), e12198; EcoMat, (2023) e12356; Advanced Materials Technologies (2023) 2201500]. The authors need to further provide these requirements.

 -, In figure 11, the scheme doesn’t provide details information. To understand a photo, a reviewer suggests the authors to modify this.

Author Response

Dear reviewer#2,

Please find the detailed responses in the attached file below and the corresponding revisions/corrections highlighted/in track changes in the re-submitted files.

Best regards,

Dr. Avetissian

Round 2

Reviewer 1 Report

Comments and Suggestions for Authors

This revision partially addresses my concerns. However, previous comments 4 and 5 still need to be supported by more specific data. Although there may be no direct experimental data in the literature under cochlear implant working conditions. However, authors still should simulate and compare the outputs of different design configurations under cochlear implant conditions based on this paper and the literature.

Author Response

Dear reviewer,

The authors greatly appreciate your time and effort in examining this manuscript. Please find the detailed response to your question in the attached file and the corresponding revisions highlighted in the re-submitted files.

Best regards,

Dr. Avetissian

Round 3

Reviewer 1 Report

Comments and Suggestions for Authors

No more comments.